# In Vitro Incorporation of *Helicobacter pylori* into *Candida albicans* Caused by Acidic pH Stress

**DOI:** 10.3390/pathogens9060489

**Published:** 2020-06-19

**Authors:** Kimberly Sánchez-Alonzo, Cristian Parra-Sepúlveda, Samuel Vega, Humberto Bernasconi, Víctor L. Campos, Carlos T. Smith, Katia Sáez, Apolinaria García-Cancino

**Affiliations:** 1Laboratory of Bacterial Pathogenicity, Department of Microbiology, Faculty of Biological Sciences, University of Concepción, Concepción 4070386, Chile; kimsanchez@udec.cl (K.S.-A.); cparras@udec.cl (C.P.-S.); samvega@udec.cl (S.V.); csmith@udec.cl (C.T.S.); 2Laboratorios Pasteur, Concepción 4030000, Chile; hbernasconi@lpasteur.cl; 3Laboratory of Environmental Microbiology, Department of Microbiology, Faculty of Biological Sciences, University of Concepcion, Concepción 4070386, Chile; vcampos@udec.cl; 4Department of Statistics, Faculty of Physical and Mathematical Sciences, University of Concepción, Concepción 4070386, Chile; ksaez@udec.cl

**Keywords:** Helicobacter pylori, Candida albicans, stress, pH, intracellular bacteria

## Abstract

Yeasts can adapt to a wide range of pH fluctuations (2 to 10), while *Helicobacter pylori*, a facultative intracellular bacterium, can adapt to a range from pH 6 to 8. This work analyzed if *H. pylori* J99 can protect itself from acidic pH by entering into *Candida albicans* ATCC 90028. Growth curves were determined for *H. pylori* and *C. albicans* at pH 3, 4, and 7. Both microorganisms were co-incubated at the same pH values, and the presence of intra-yeast bacteria was evaluated. Intra-yeast bacteria-like bodies were detected using wet mounting, and intra-yeast binding of anti-*H. pylori* antibodies was detected using immunofluorescence. The presence of the *H. pylori* rDNA 16S gene in total DNA from yeasts was demonstrated after PCR amplification. *H. pylori* showed larger death percentages at pH 3 and 4 than at pH 7. On the contrary, the viability of the yeast was not affected by any of the pHs evaluated. *H. pylori* entered into *C. albicans* at all the pH values assayed but to a greater extent at unfavorable pH values (pH 3 or 4, *p =* 0.014 and *p =* 0.001, respectively). In conclusion, it is possible to suggest that *H. pylori* can shelter itself within *C. albicans* under unfavorable pH conditions.

## 1. Introduction

There is evidence showing that certain bacteria can become permanent or transitory endosymbionts of eukaryotic cells. This condition may have been acquired during evolution in order to serve as a specialized niche in which bacteria are protected from environmental stress and their transmission to a new host is facilitated [1,2,3]. Therefore, the endosymbiotic relationship between bacteria and hosts is a conserved phenomenon and has an important impact on the evolution of microorganisms [4]. For example, yeasts are highly sophisticated microorganisms with a notable capacity of change, which can adapt to environmental stress [5] and establish symbiotic relationships with certain microorganisms [6]. Furthermore, it has been found that *Candida albicans* may harbor *Helicobacter pylori* cells [3,6,7,8,9,10].

*Candida* is considered to be a genus of opportunistic pathogenic yeasts causing infection in immunocompromised individuals or in those with an altered health status, such as individuals with diabetes [11]. These yeasts can also cause infections in healthy individuals, such as women of fertile age [12,13]. Several *Candida* species share the ability to grow under diverse environmental conditions, including pH ranges from 2 to 10 in different anatomical and environmental niches, even when the availability of nutrients is restricted [14,15]. *C. albicans* is mainly associated with humans, growing on the surface of the mucosae of the gastrointestinal and genitourinary tracts and on the skin [16,17,18].

On the other hand, *H. pylori* is a facultative intracellular pathogenic microorganism. Some authors report that *H. pylori* may take advantage of this characteristic to protect itself from factors negatively affecting its viability [2,19]. The association between *H. pylori* and *Candida* was first proposed in 1998, when yeast colonies were found as contaminants of gastric biopsies in blood agar plates [20]. Optical microscopy showed the presence of bacteria-like bodies (BLBs) rapidly moving within the vacuoles of these gastric yeasts, which were purified and identified as *Candida*. Polymerase chain reaction (PCR) was used to reveal the bacterial nature of these BLBs [20]. Optical microscopy as well as the detection of *H. pylori*-specific genes and the immunodetection of proteins of this bacterium in yeasts of the genus *Candida* confirmed the intra-yeast presence of the pathogen *H. pylori*. These results supported the idea that *C. albicans* might act as a reservoir of *H. pylori* outside the human stomach [7,8,9,10,20,21,22]. Other authors reported the coexistence of *C. albicans* and *H. pylori* has a synergistic effect on the pathogenesis of giant gastric ulcers [23]. Further knowledge concerning the details of the interactions between these two microorganisms will eventually provide important information for the treatment of infectious diseases caused by *H. pylori*, particularly those showing resistance to antimicrobial agents [24].

*H. pylori* infection is directly related to the development of gastric pathologies, such as chronic gastritis, peptic ulcers, mucosa-associated lymphoid tissue lymphoma (MALT lymphoma), and gastric cancer [25], justifying the interest from many researchers to improve our knowledge regarding this pathogen. A further reason to study this microorganism is the fact that its transmission routes are not yet fully known. In general, it is accepted that the most probable *H. pylori* transmission routes are the fecal–oral, oral–oral, or gastro–oral ones [26]. Intrafamily dissemination has been proposed as the most common form to acquire the infection, the mother being the main factor participating in the dissemination of this pathogen. This asseveration is based on an investigation by Burucoa and Axon [27], in which, after studying children under 15 years of age whose parents were *H. pylori* positive, they found 10 children were infected with a strain whose genotype was the same as that of the mother, while only 2 children shared the genotype of the microorganism found in the father [28].

Moreover, it has been proposed that a concurrence of *H. pylori* and *C. albicans* indicates a more intimate relationship, which protects *H. pylori* from environmental stress [20]. Thus, it is necessary to evaluate the different factors that might favor the harboring of H. pylori within yeasts. One of these factors is the role of acidic pH on the relationship between both microorganisms. Therefore, the aim of this work was to evaluate, in vitro, the role of acidic pH as a trigger for the internalization of H. pylori into the vacuole of C. albicans yeast cells.

## 2. Results

### 2.1. Acidic pH Stress Factor Assay

This assay allowed us to determine the viability of *H. pylori* and *C. albicans* exposed to different pH values. When exposed to pH 3, *H. pylori* showed an abrupt decrease in bacterial counts during the first hour of exposure. When exposed to pH 4, a gradual decrease in the bacterial counts was observed until 4 h; the counts then remained unchanged until 5 h, which was significantly different to the control (pH 7). Regarding *H. pylori* growth at pH 7, no changes in the bacterial cell counts were observed during the 5 h of the assay (Figure 1). Regarding the growth of *C. albicans*, under pH stress conditions, its growth at pH 3 was slower, not reaching the stationary phase even after 50 h of incubation (Figure 2).

### 2.2. In Vitro Entry of H. pylori J99 into C. albicans ATCC 90028

Regarding the entry of *H. pylori* into *C. albicans,* the first BLBs were observed as dots whose movement was restricted to the size of the vacuole during the first hour of co-culturing, as observed in (Figure 3). The immunofluorescent assay allowed us to identify intracellular *H. pylori* within yeasts (Figure 4). The percentage of *C. albicans* harboring bacteria observed by optical microscopy at different times and pH values are shown in Table 1. There was a significant difference in yeast harboring *H. pylori* at pH 3 or pH 4 when compared to pH 7, and at pH 4, the number of yeast harboring *H. pylori* was constant through time (Table 1). No significant difference regarding the entry of bacteria was observed when comparing pH 3 and pH 4, but the entry of bacteria into *C. albicans* at these two pH values showed a significant difference when compared to pH 7 at 24 and 48 h. The percentage of BLBs harboring yeasts showed variations according to the time of exposure to the stressing factor (pH); there were higher percentages of yeasts harboring BLBs during the first and last hour that the co-cultures were incubated at acidic pH values, showing a greater significance at pH 4 (*p* = 0.001) than at pH 3 (*p* = 0.014).

### 2.3. Amplification of The H. pylori J99 16S rDNA Gene from the Total DNA Extracted from C. albicans ATCC 90028

After agarose gel electrophoresis, it was possible to observe the presence of a band of 110 bp in the total DNA of bacterium yeast co-cultures incubated at pH 7, 4, and 3 (Figure 5, lanes 1–3) and also present in the positive *H. pylori* J99 control, but absent in the yeast not co-cultured with the bacterium. This band is compatible with the *16S rDNA* gene of *H. pylori* in the total DNA extracted from *C. albicans* ATCC 90028 co-cultured with the bacteria at the three different pH values. This result confirmed the above-mentioned microscopic observations indicating the presence of *H. pylori* within *C. albicans* cells.

## 3. Discussion

It is known that the ecological niche of *H. pylori* is the human stomach, an acidic environment (pH 2.0 to 3.0) to which this bacterium has adapted. However, some *H. pylori* genes have also been reported in water (pH 6.5–7.4), milk (pH 6.6–6.8), and in the human oral cavity (pH 5.5–7.4), but viable bacteria have so far not been cultured from these samples. It is possible that the *H. pylori* strains present in these environments are dormant (i.e., viable but non-culturable). This assumption is based on the fact that *H. pylori* in drinking water or in the presence of stressing factors may acquire a coccoid morphology, associated with a state of dormancy [29,30].

Our study showed that *H. pylori* reduced its viability over time in an acidic pH, particularly pH 3, the lowest pH value analyzed (Figure 1). This is due to the absence of urea in the medium because, in the gastric environment, this bacterium neutralizes the pH through secretion of the urease enzyme that hydrolyzes urea to produce ammonium and CO_2_ [31]. In the absence of urea, this bacterium cannot raise the environmental pH and must seek other strategies to protect itself from stressors, such as its ability to invade and remain viable within eukaryotic cells. *H. pylori* has been reported not only to invade and survive within human cells, but also within free-living amoebas and yeasts isolated from the mouth of adults and newborns, as well as the vagina, yogurt, honey, and other environmental sources [2,19,32,33,34] The results of these studies suggest that eukaryotic organisms might protect *H. pylori* and be a transmission vehicle for this bacterium. In addition, *C. albicans* has a wide range of tolerance to pH, which was confirmed in this study: the *C. albicans* ATCC 90028 strain was not affected by the acidic pH values assayed (Figure 2), not reaching the stationary phase even after 50 h of incubation. Fungi have the capacity to adapt to or modify the environmental pH, secreting acids or alkalis, with some pathogenic fungi even secreting acids as a strategy to damage the tissues of the host [35]. Therefore, the wide range of pH tolerance showed by the *C. albicans* ATCC 90028 strain was expected.

When *H. pylori* and *C. albicans* were co-cultured at an acidic pH, the percentage of yeasts harboring bacteria was nearly twice that observed in the control (pH 7), demonstrating that although *H. pylori* enters the yeast without the presence of an apparent stressor, the pH is a factor capable of influencing the entry of *H. pylori* into yeasts. Regarding the difference between the yeasts harboring *H. pylori* at pH 4 and pH 3, a larger percentage of *C. albicans* harboring bacteria would be expected at lower pHs; nevertheless, our results showed the opposite. The cell wall of yeasts is normally composed of two layers, an external one composed of mannoproteins, called the fibrillar layer, and an inner layer composed of chitin and glucans [15]. Sherrington et al. [15] demonstrated that *C*. *albicans* exposed to acidic pHs (pH 2, 4, and 6) modified the structure of their cell wall, causing a significant loss of the fibrillar layer, and increasing the exposure of chitin and β-glucans at pH values below 4, as compared to those cultured at pH 8 or in yeast extract peptone dextrose (YPD) medium at pH 7 [36]. Therefore, it is possible to postulate that this change in the cell wall of the yeast might impede the entry of *H. pylori* at pH values below 4. It has also been reported that changes in pH also induce the remodeling of the plasmatic membrane of fungi. These modifications include changes in the concentration of fatty acids and ergosterol, turning the membrane into a more rigid structure [37,38]. Thus, this is another factor which may affect the entry of *H. pylori* into yeasts.

In addition, regarding the issue of *H. pylori* localization within vacuoles of eukaryotic cells, Siavoshi et al. were able to identify the presence of this bacterium within vacuoles of yeasts—the organelle where other viable bacterial cells were also observed [2,39]. It is known that the vacuole of yeasts plays important roles, including pathogenicity. It has been described that mutant *C. albicans* with severe defects for the biogenesis of vacuoles are avirulent, i.e., not able to form hyphae, which, for these fungi, are essential structures to invade and for its pathogenicity [40]. Other functions attributed to the vacuole are the classification of signals, osmoregulation, and storage of metabolites, such as Ca^2+^, phosphate, and amino acids [41]. Phosphate and amino acids may offer a sustaining environment for *H. pylori* within this yeast.

Although other authors have reported the presence of *H. pylori* within vacuoles of yeasts obtained from different sources, there is one study that proposes a stressing factor promotes the exit of intracellular bacteria from yeasts [6]. The present work is, to the best of our knowledge, the first report on the factors involved in the entry of *H. pylori* into yeasts, demonstrating that acidic pH values affect the entry of *H. pylori* into *C. albicans*. Since scientific reports with this focus are scarce, the results here described are of great importance for future research.

## 4. Materials and Methods

### 4.1. Strains and Culture Conditions

This work was done using the reference strain *H. pylori* J99 (ATCC 700824, urease+*cagA*+*vac*A+, isolated from a duodenal ulcer) and the *C. albicans* ATCC 90028 strain. *H. pylori* J99, used as a representative of the species *H. pylori,* was cultured in plates containing Columbia agar (OXOID, United Kingdom) supplemented with 5% horse blood and DENT (OXOID, Basingstoke, United Kingdom) at 37 °C for 72 h in an incubator (Thermo Scientific, Waltham, MA, USA) under microaerobic conditions (10% CO_2_). *C. albicans* ATCC 90028 was used as a representative of the genus *Candida*. It was cultured on Sabouraud agar (Merck, Darmstadt, Germany) supplemented with chloramphenicol (OXOID, Basingstoke, United Kingdom) and incubated at 37 °C for 24 h in an incubator (ZHICHENG, Shanghai, China) under aerobic conditions. *H. pylori* J99 strain is part of the culture collection of the Laboratory of Bacterial Pathogenicity of the University of Concepcion, Chile, the premises where this work was done. *C. albicans* ATCC 90028 was donated by Dr. Patricio Godoy, Institute of Clinical Microbiology, Universidad Austral de Chile, Valdivia, Chile. 

### 4.2. Exposure to Acidic pH Stress Assay

For this assay, *H. pylori* J99 and *C. albicans* ATCC 90028 were independently exposed to pH 3, 4, and 7 in order to choose a pH that did not affect *C. albicans* viability, but generated stress for *H. pylori* (pH 3 and 4). Cultures at pH 7 were used as controls. Firstly, both microorganisms were cultured until each culture reached its exponential phase. Then, each strain was independently cultured in 55 mL of Brain Heart Infusion (BHI) medium (Difco, Wokingham, United Kingdom) supplemented with 1% yeast extract, and the appropriate pH values were obtained using 10 M hydrochloric acid (HCl) or 10 M sodium hydroxide (NaOH) and a pH meter (Bante, Shanghai, China); the initial inoculum was adjusted to a turbidity equivalent to McFarland 7 (2 × 10^9^ colony-forming units (CFU)/mL) and incubated for 72 h at 37 °C under microaerobic conditions. Every 4 h, a 200 µL aliquot was obtained from each culture and transferred to a well of 96-well plates, and the absorbance was measured spectrophotometrically (TECAN, Männedorf, Switzerland) at a wavelength of 600 nm, as reported by Huang et al. [30]. In parallel, microscopic observations (wet mount and Gram staining) were performed for each 4 h sampling time in order to observe the morphology of each microorganism (*H. pylori* J99 and *C. albicans* 90028). In addition, every 4 h, *H. pylori* bacterial counts were determined using the microdrop technique, wherein 100 µL of *H. pylori* culture was diluted with 900 µL saline solution, and five serial dilutions were prepared. Thereafter, 10 µL of the original dilution and of the five serial dilutions were seeded, by triplicate, in plates containing Columbia agar supplemented with DENT (OXOID, United Kingdom) and incubated at 37 °C for 36 h under microaerobic conditions. The number of CFUs was counted, and the results were expressed as CFU/mL using the formula shown below [30]. This assay was done in triplicate.
$$\mathrm{CFU}/\mathrm{mL}=\frac{\mathrm{average}\;\mathrm{of}\;\mathrm{counted}\;\mathrm{CFU}\;\times \;\mathrm{dilution}\;\mathrm{factor}\;}{\mathrm{seeded}\;\mathrm{volume}\;\mathrm{expressed}\;\mathrm{in}\;\mathrm{mL}}$$


### 4.3. In Vitro Assessment of H. pylori J99 Entry into C. albicans ATCC 90028 Caused by Acidic pH

*H. pylori* J99 and *C. albicans* ATCC 90028 were co-cultured as described in the following. A 1 × 10^5^ CFU/mL *H. pylori* suspension was prepared from a 36 h culture. In parallel, a 1 × 10^3^ CFU/mL *C. albicans* ATCC 90028 suspension was prepared from a 20 h culture. Both suspensions were prepared in 15 mL Falcon tubes containing Brain Heart Infusion (BHI) broth supplemented with 1% yeast extract. Then, 100 µL of *H. pylori* suspension and 500 µL of *C. albicans* suspension were mixed in a Falcon tube containing 15 mL BHI broth supplemented with 1% yeast extract and incubated at 37 °C under microaerobic conditions. Aliquots of 20 µL of the co-culture were obtained at 0, 1, 3, 6, 12, 24, and 48 h and observed as wet mounts using immersion oil and the 100X objective lens of an optical microscope (Leica, Wetzlar, Germany) to search for the presence of yeasts harboring bacterium-like bodies (BLBs). The percentage of BLBs carrying yeasts was calculated after counting the number of yeasts harboring BLBs or not harboring them in 100 microscopic fields.

In order to have as many yeast cells harboring *H. pylori* as possible, the time with the higher percentage of intra-yeast BLBs for each pH assayed was selected for the following assays. Extracellular bacteria were eliminated as reported by Moreno-Mesonero et al. (2016) [19], with modifications. Each Falcon tube containing the co-culture was centrifuged at 5000× *g* for 3 min, the pellet was resuspended using 200 µL of 104 ppm sodium hypochlorite, and then it was incubated for 1 h to kill extracellular bacteria. Then, the suspension was centrifuged at 5000× *g* for 3 min, and the cells were washed thrice using 1.5 mL phosphate-buffered saline (PBS) and the same centrifugation protocol. Then, the collected yeasts were seeded on Sabouraud agar (Merck, Darmstadt, Germany) supplemented with chloramphenicol (OXOID, Basingstoke, United Kingdom) and incubated at 37 °C for 24 h in an incubator (ZHICHENG, Shanghai, China) under aerobic conditions. Next, three reseedings were done to eliminate possible persistent extracellular bacteria. At each reseeding, wet mounts were prepared and analyzed using an optical microscope to confirm the presence of intra-yeast BLBs. This assay was done in triplicate.

### 4.4. Detection of H. pylori J99 Entry into C. albicans ATCC 90028 by Immunofluorescence

This assay was done using Fluorescein Isothiocyanate (FITC) (Abcam, Cambrige, United Kingdom)—labeled rabbit polyclonal anti-*H. pylori* IgG antibodies at a concentration of 5 mg/mL (ab30954). *C. albicans* ATCC 90028 and *H. pylori* J99 strains were used as negative and positive controls, respectively. Samples and controls were diluted in Eppendorf tubes containing 1 mL PBS 1× and adjusted to a turbidity equivalent to McFarland 2 (6 × 10^8^ UFC/mL). Using 96-well plates, 200 μL of each sample or control was placed in a well, and 1 μL of FITC-labeled anti-*H. pylori* IgG was added and incubated for 1 h at room temperature in darkness. After this time span, 10 μL was obtained from each well and placed on a slide observed at a wavelength of 528 nm using a confocal microscope (LSM780 NLO, ZEISS), with an Ar488 nm excitation and 490–560 nm emission. Transmitted light images, corresponding to fluorescence images with a 2–4 μm thickness, were obtained using the Zen 2012 software. 

### 4.5. Amplification of The H. pylori 16S rDNA Gene from The Total DNA Extracted from C. albicans ATCC 90028

Extraction of the total DNA of *C. albicans* ATCC 90028 was done using the UltraClean Microbial DNA Isolation kit (M.O. BIO, Carlsbad, CA, USA), following the instructions of the manufacturer, and the extracted DNA was quantified by spectrophotometry (TECAN, Männedorf, Switzerland). The *16S rDNA* gene of *H. pylori* was amplified using the SapphireAmptf Fast PCR Master Mix kit (TAKARA BIO INC, Japan). For each sample, 12.5 μL of Master Mix, 1 μL of the forward primer F- 5′CTC GAG AGA CTA AGC CCT CC 3′, 1 μL of the reverse primer R- 5′ATT ACT GAC GCT GAT GTG C 3′,5 μL of DNA of the sample, and 5.5 μL of PCR-grade water were added to obtain a final volume of 25 μL of PCR mixture. PCR conditions were as follows: initial denaturation 94 °C/1 min, denaturation temperature 98 °C/5 s, hybridization temperature 53 °C/5 s, and extension at 72 °C/40 s. Thirty cycles for each PCR reaction were done using a thermocycler (Eppendorf, Hauppauge, NY, USA). Amplification of the *16S rDNA* gene was confirmed after 2% agarose gel electrophoresis (Lonza, Walkersville, MD, USA) run at 80 V for 90 min, and the gels were visualized under UV light using an Enduro model transilluminator (Labnet, Edison, NJ, USA) [22].

### 4.6. Statistical Analysis

To determine if there was a statistically significant difference in the percentage of yeasts harboring *H. pylori* at the three pH values assayed, the Chi-squared test was applied. *p*-values < 0.05 were considered as significant. Data were processed using the SPSS 24.0 software (IBM Company, Armonk, NY, USA).

## 5. Conclusions

The results suggest that *H. pylori* can shelter itself within *C. albicans* under unfavorable pH conditions.

## Figures and Tables

**Figure 1 pathogens-09-00489-f001:**
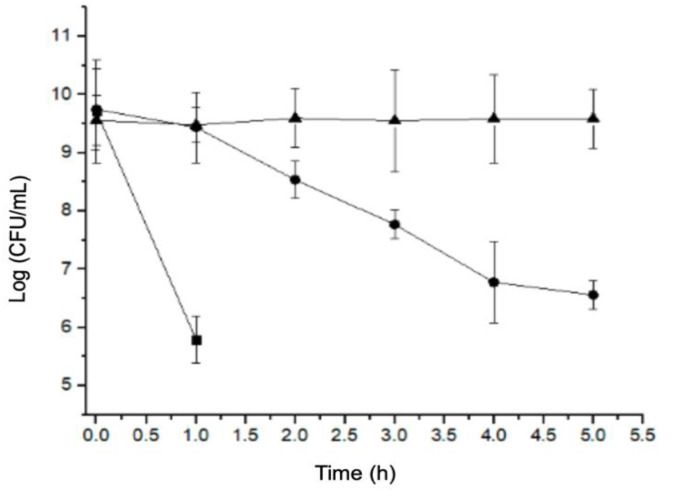
Growth curve of *Helicobacter pylori* J99 at pH 3 (■), pH 4 (●), and pH 7 (▲).

**Figure 2 pathogens-09-00489-f002:**
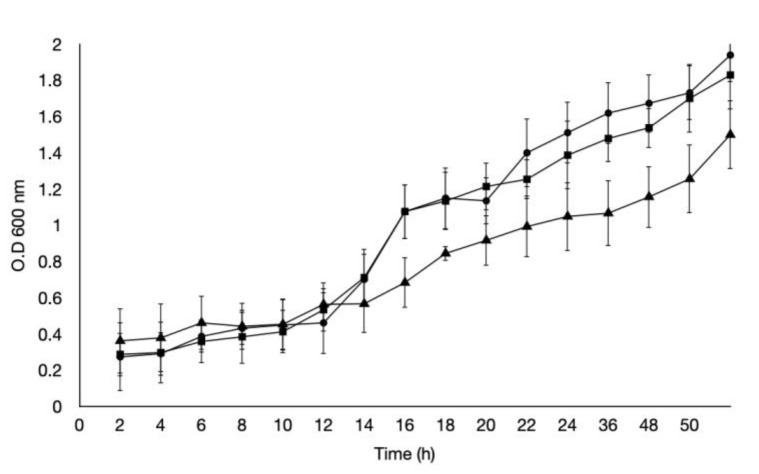
Growth curve of *Candida albicans* at pH 3 (■), pH 4 (●), and pH 7 (▲). In vitro entry of *H. pylori* J99 into *C. albicans* ATCC 90028.

**Figure 3 pathogens-09-00489-f003:**
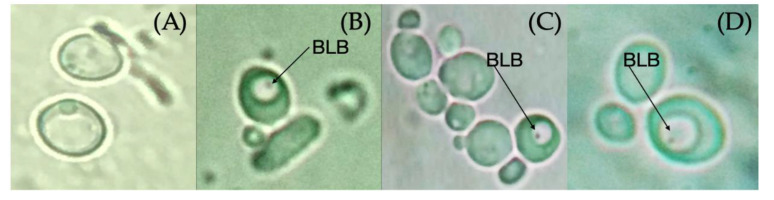
Bacterium-like bodies (BLBs) within *C. albicans* ATCC 90028 vacuoles after 48 h of incubation at pH 3, pH 4, and pH 7, observed by optical microscopy using a 100× objective lens. (**A**) Negative control, *C. albicans* 90028 strain; (**B**) incubation at pH 3; (**C**) incubation at pH 4; and (**D**) incubation at pH 7.

**Figure 4 pathogens-09-00489-f004:**
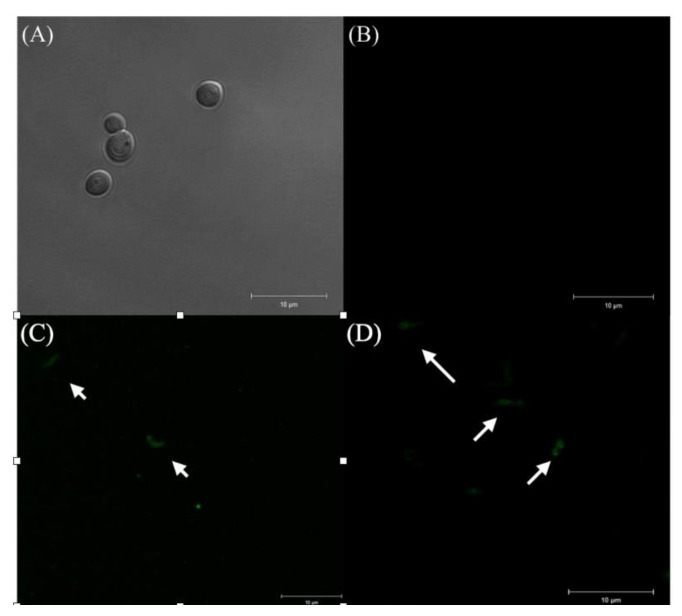
Immunofluorescent assay using Fluorescein Isothiocyanate (FITC)-labeled anti-*H. pylori* IgG polyclonal antibodies. (**A**) *C. albicans* ATCC 90028 not co-cultured with the bacterium (negative control) observed by differential interference contrast (DIC); (**B**) absence of fluorescence in the negative control; (**C**) presence of fluorescence emitted by *H. pylori* J99 (white arrows) (positive control); and (**D**) fluorescence emitted by the intracellular presence of *H. pylori* J99 (white arrows) in yeasts.

**Figure 5 pathogens-09-00489-f005:**
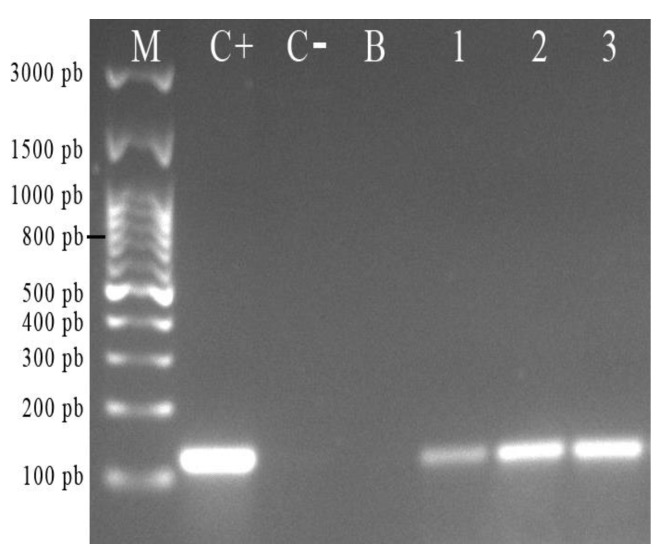
Detection, in 2% agarose gel, of the *16S rDNA* gene of *H. pylori* J99 in the total DNA extracted from *C. albicans* ATCC 90028 previously co-incubated with the bacterium for 48 h at pH 3, pH 4, and pH 7. (**M**) Molecular weight marker; (**C^+^**) positive control (*H. pylori* J99); (**C^−^**) negative control (*C. albicans* 90028 incubated in the absence of the bacterium); (**B**) blank (polymerase chain reaction (PCR) degree water); (**1**–**3**) amplicons from the total DNA of co-cultures incubated at pH 7, 4, and 3, respectively.

**Table 1 pathogens-09-00489-t001:** Percentage of yeasts harboring bacterium-like bodies (BLBs) within *C. albicans* ATCC 90028 at different times of incubation at pH 3, pH 4, and pH 7, observed by optical microscopy.

Time (h)	pH 3	pH 4	pH 7	*p*-Value
0	-	-	-	
1	11.7%	13.7%	6.3%	0.0559
3	3.0%	3.3%	2.7%	0.9540
6	4.2%	3.3%	2.2%	0.5773
12	7.1%	6.4%	2.2%	0.0963
24	8.7%	12.5%	3.3%	0.0031
48	11.1%	18.7%	4.1%	0.0309

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
