# Peer review of "In Vitro Incorporation of Helicobacter pylori into Candida albicans Caused by Acidic pH Stress"

_pathogens, 2020, doi:10.3390/pathogens9060489_

Round 1

Reviewer 1 Report

The aim of the study by Sanchez-Alonzo et al., entitled "In vitro incorporation of Helicobacter pylori into Candida albicans caused by acidic pH stress", was to evaluate co-operation of bacteria and yeast in order to cope stressful condition, which in a small part imitating those occurring during infection with H.p.

The research carried out by the authors seemed to me interesting, however, this aspect of the pathogenesis of H.p. has been studied for many years, and there is still no common agreement as to their interpretation. Therefore, the authors should particularly emphasize the new approach to the problem and underline the obtained original information that goes beyond the earlier available reports. One could expect more universal knowledge than studying the impact of something on something without assessing and describing the proven mechanisms of the observed phenomenon.

General comments

The current MS version is not devoid of some limitations. The authors should consider the following specific comments.

Introduction section (1.) presents and justifies the purpose of research in very general terms. There is no so-called "fluidity" of the text. Many repetitions of the same content are noted, so the content should be re-written. Also for example, the first paragraph on page 1 contains data not related to the purpose of the research, thus it should be shortened or deleted.

In general, the so-called guiding idea of the study is missing, which should be an essential part of any original study report.

Material and Methods section (4.)

The current description of the research methods used does not allow them to be repeated in another laboratory, which is an essential condition for their reliability. They should necessarily be described in a more accurate version.

Questions, for example:

- How many times each experiment was performed and how many repetitions of individual trials were done.

Subsection Strains and culture conditions

- characteristics of H.p. J99 strain (clinical?) should be presented: at least urease, VacA, CagA production

Subsection In vitro assessment of H. p. J99 entry to ....

-First paragraph: question - is the 40x lens sufficient to see intracellularly located bacteria?

-Second paragraph, line 2 what for ..."showing the best internalization of H. pylori cells within C. albicans ATCC 90028 yeast cells was chosen:.

- How is possible using 15 ml Falcon tubes filled them with ... "200 ml 1104 (?) ppm sodium hypochlorite"...

- Precise justification of sodium hypochlorite using is needed - only to kill H.p. located extracellularly? What did C.a. cells look like after this stage? Or maybe this stage caused the fixation and permeabilization of yeast cells. Detailed description and justification is needed. References for such a protocol would be necessary.

- Data on the antibodies used, their dilution and the immunological "staining" protocol are necessary. In a previous publication by team members (ref. 31. Matamala

-Valdés et al., 208) stated that for this purpose they used rabbit polyclonal IgG anti- H. pylori antibodies marked with FITC, whose concentration was 5,000 mg/ml, which is a clear error).

Results section (2.)

Presentation of the results is weak. Currently, I do not undertake to submit substantive comments. The MS version presented is chaotic, and therefore difficult to analyze. This part should be rewritten after correcting the description of the Materials and Methods section. There is no need to describe in detail the results shown in the Tables and Figures.

Comments:

- Subsection In vitro entry of H.p. J99 into ......How looks like control C. a. cells - this is not shown in Fig. 3.; at what magnification of the lens was then taken photos -100x (legend to Fig. 3) or 40x as given in Mat & Meth.

-legend to Fig. 4. - please correct fluoresce to flluorescence

-Table 1. standard deviation o the presented percentages as well as statistical significance between data groups should be completed

Discussion section (3)

This part is definitely too long. It contains many repetitions and the same information (epidemiological) as in the Introduction, mostly not necessarily related to the purpose of the study.

Moreover, in many places the interpretation of the important observations made is missing or copied from well known publications. It should be re-edited to provide more constructive, scientifically proven information.

Some of valuable publications from the last years have been omitted.

- The authors should skip first paragraph

- 5. paragraph is a repetition of content from the previous page

- 9. paragraph is not needed

Conclusions section (5.)

This statement is not entirely justified because it mainly contains knowledge that is known, and few other accurate data is obtained during the study.

Authors should convince readers of the originality of their observations.

On the one hand, the authors join research carried out in this area by several teams, on the other hand, they do not attempt to deepen knowledge in the above field and do not provide new observations.

Their declarations of intent to conduct further research are so general that they do not provide any scientific basis for expanding current knowledge.

Author Response

Dear Reviewer:

I appreciate all your comments and contributions to the document, our responses are attached to the document.

Best regards,

Reviewer 2 Report

The manuscript entitled “In vitro incorporation of Helicobacter pylori into Candida albicans caused by acidic pH stress” have determined the different factors (stress factors) which might favor the entry H. pylori to yeast cell. This is an important topic because, both microorganisms influence on human health. Yeast C. albicans are commensal microorganisms that grows on the skin and mucosal surfaces of healthy individuals. However, in some cases are associated with opportunistic infection in both animals and humans, especially in immunologically weak and immunocompromised people such as those with HIV/AIDS. In susceptible patients, C. albicans can enter the bloodstream by translocation across the mucosa of the gastrointestinal tract.

Authors should correct manuscript according to the suggestion and completed some information.

Minor issues:

Introduction:

Please clarification aim of the study. The introduction lacks a clearly defined aim of study. The Authors described C. albicans and H. pylori but did not explain possible relationships between microorganisms. Are there any epidemiological data suggesting interaction between them? I think that relationship exists.

Results:

Figure 2, standard deviation are missing

Materials and Methods:

co-cultured condition (inoculum size) of yeast and bacteria was optimized?

References:

Authors should checked and corrected Reference no 22, 39, 56 and 60 according to journal guidelines

Author Response

(The authors gave the same response as above.)

Reviewer 3 Report

The paper entitled „In vitro incorporation of Helicobacter pylori into Candida albicans caused by acidic pH stress“ submitted to Pathogens by Sanchez-Alonzo et al. attracted my interest. The authors nicely described the capacity of H. pylori to adapt to low pH and described the possibility of H. pylori to enter C. albicans as a possible way how to protect itself from unhostile environment.

I am wondering how the low pH impact the growth curve of C. albicans. Actually, the manuscript would profit from data demonstrating the effect of the entrance of H. pylori into C. albicans on the growth capacity. The authors suggested that H. pylori survives in BLBs (Shavoshi et al, 2019) but there are data missing how this „sheltering“ impact the viability/growth curve of H. pylori under different pH.

Minor comments:

Introduction

Last sentence of the first paragraph is confusing – probably replacement of IF by OF should solve it: Burucoa and Axon, also studied children below 15 years of age whose parents were positive for H. pylori and reported that 10 children showed to be infected with a strain whose genotype was the same as to the one of the mother while only two children shared the genotype if the microorganism causing the infection with their fathers. Change to: Burucoa and Axon, also studied children below 15 years of age whose parents were positive for H. pylori and reported that 10 children showed to be infected with a strain whose genotype was the same as to the one of the mother while only two children shared the genotype of the microorganism causing the infection with their fathers.

Discussion

Last paragraph: replace differential by different or distinct

Methods:

Title of the third paragraph – „n“ is missing in C. albicans

The first sentence of the section: Detection of H. pylori J99 entry into C. albicans ATCC 90028 by immunofluorescence

„l“ is missing in labelled

Funding:

Point is missing at the end of the sentence.

Author Response

(The authors gave the same response as above.)

Round 2

Reviewer 1 Report

I accept most of the amendments and additions made by the authors, which have undoubtedly raised the scientific value of MS.

However, because I want to help authors, I think that some minor modifications to the text are still needed, which will additionally support this.

1). If it is possible, MS should be language tested by a native speaker.

2). Minor changes in the content of Introduction would be indicated:

- L 46 .... "It has been found that C. albicans may harbor Helicobacter pylori cells". should be moved after L 38.

- Next will be L 39-45; then - L 46 - On the other hand,  ......H. pylori is a facultative intracellular pathogenic microorganism). up to L 60.

- L 61-75 - it is suggested to consider the content of this phrase and / or modify it. There is no clear connection with earlier and later (L 76-81) threads.

3) Results

- L 97- 102 - the same information is repeated twice.

- L 102-110 - the comments on the results presented are complex, and therefore difficult to understand by the potential reader. They should be modified to make them clearer.

4) Discussion

- L 141 -144 - should be deleted - this information is not relevant to the context of the study.

5) Materials and Methods

- L 216 - .... "as described in section Strains and culture conditions"... should be deleted

- L 217 - word: "independently" .. should be deleted

Reviewer 3 Report

The revised version of the manuscript submitted by Sanchez-Alonzo et al. is substantially improved. Nevertheless, there are several minor issues which should be addressed.

Figure 1 use CFU instead of UFC and explain the abbreviation in figure legend 

Figure 2 Y axis - use O.D. instead of D.O. and explain the abbreviation in figure legend.

Fig. 5 use PCR-grade water instead of degree (line 138)
